# Epigenetic and Transcriptomic Programming of HSC Quiescence Signaling in Large for Gestational Age Neonates

**DOI:** 10.3390/ijms23137323

**Published:** 2022-06-30

**Authors:** Alexandre Pelletier, Arnaud Carrier, Yongmei Zhao, Mickaël Canouil, Mehdi Derhourhi, Emmanuelle Durand, Lionel Berberian-Ferrato, John Greally, Francine Hughes, Philippe Froguel, Amélie Bonnefond, Fabien Delahaye

**Affiliations:** 1Inserm U1283, CNRS UMR 8199, European Genomic Institute for Diabetes, Institut Pasteur de Lille, 59000 Lille, France; alexandre.pelletier.etu@univ-lille.fr (A.P.); arnaud.carrier@pasteur-lille.fr (A.C.); mickael.canouil@cnrs.fr (M.C.); mehdi.derhourhi@cnrs.fr (M.D.); emmanuelle.durand@cnrs.fr (E.D.); lionel.berberian@cnrs.fr (L.B.-F.); amelie.bonnefond@cnrs.fr (A.B.); 2Lille University Hospital, University of Lille, 59000 Lille, France; 3Department of Obstetrics & Gynecology and Women’s Health, Albert Einstein College of Medicine, 1300 Morris Park Ave, Bronx, NY 10461, USA; yongmei.zhao@einsteinmed.org; 4Department of Genetics, Albert Einstein College of Medicine, 1301 Morris Park Avenue, Price Building, Room 322, Bronx, NY 10461, USA; john.greally@einsteinmed.org; 5Obstetrics & Gynecology and Women’s Health, Division of Maternal-Fetal Medicine, Montefiore Medical Center, Albert Einstein College of Medicine, Bronx, NY 10461, USA; fhughes@montefiore.org; 6Department of Metabolism, Digestion and Reproduction, Imperial College London, Exhibition Rd, South Kensington, London SW7 2BX, UK

**Keywords:** epigenomics, single-cell, stem-cells, fetal programming, hematopoiesis

## Abstract

Excessive fetal growth is associated with DNA methylation alterations in human hematopoietic stem and progenitor cells (HSPC), but their functional impact remains elusive. We implemented an integrative analysis combining single-cell epigenomics, single-cell transcriptomics, and in vitro analyses to functionally link DNA methylation changes to putative alterations of HSPC functions. We showed in hematopoietic stem cells (HSC) from large for gestational age neonates that both DNA hypermethylation and chromatin rearrangements target a specific network of transcription factors known to sustain stem cell quiescence. In parallel, we found a decreased expression of key genes regulating HSC differentiation including *EGR1*, *KLF2, SOCS3,* and *JUNB*. Our functional analyses showed that this epigenetic programming was associated with a decreased ability for HSCs to remain quiescent. Taken together, our multimodal approach using single-cell (epi)genomics showed that human fetal overgrowth affects hematopoietic stem cells’ quiescence signaling via epigenetic programming.

## 1. Introduction

Hematopoietic stem cells (HSC) are involved in essential processes such as inflammation, cardiovascular repair, and immunity throughout the entire lifespan [1,2]. Thus, alterations in HSC’s ability to self-renew and to adequately produce differentiated progeny have been suggested to contribute to the onset and progression of age-related diseases such as cancer and cardiovascular diseases [3,4]. Systemic alterations or the action of various stressors like aging [5,6] can result in alteration of HSC destiny, and ultimately hematopoietic functions. The early mechanisms that control their long-term functions in humans are not well understood, in part due to the diversity of phenotypes and behaviors of HSCs [1].

In mice, a maternal high-fat diet during gestation limits fetal hematopoietic stem and progenitor cells (HSPC) expansion and ability to repopulate while inducing myeloid-biased differentiation [2]. In humans, a limited number of studies have been conducted. Fetal growth was shown to alter the number of circulating CD34+ HSCs [7,8]. We previously described a global increase of DNA methylation in cord blood-derived CD34+ HSPCs from large for gestational age (LGA) infants compared to neonates with normal birth weight [3]. Still, the functional impacts of these early epigenetic alterations remain to be elucidated. Such an effort is essential to determine how these epigenetic modifications could mediate the association between early-life exposures and the induction of persistent life-long functional changes within the hematopoietic system.

We conducted a multimodal analysis combining single-cell epigenomics, single-cell transcriptomics, and in vitro analyses to link the DNA methylation alterations observed in LGA neonates with functional alterations in human cord blood-derived HSPCs. We developed novel analytical approaches to improve the integration of epigenomic and transcriptomic data. We found that the DNA hyper-methylation observed in LGA HSPC is associated with an HSC-specific decreased chromatin accessibility and gene expression of key genes involved in the HSC quiescence signaling as well as an alteration of the HSC colony-forming capacity.

## 2. Results

### 2.1. Optimized Methylation Gene Set Analysis Reveals Association between LGA DNA Hypermethylation and Stem Cell Differentiation Pathways

To confirm the LGA-associated DNA hypermethylation we previously observed, we significantly increased the power of our analysis. We expanded our original study through additional patient inclusions, thereby doubling the size of our cohort [3]. Using the HELP-tagging assay (HpaII tiny fragment Enrichment by Ligation-mediated PCR), we generated genome-wide DNA methylation data on 16 CTRL and 16 LGA cord-blood derived human CD34+ HSPC samples. We independently retrieved in this new dataset the global DNA hypermethylation initially found in LGA compared to controls [3] (Figure 1A). Then, to increase our detection power, we pooled both datasets and detected a total of 4815 differentially methylated CpGs (DMC) with 4787 CpGs hypermethylated and 28 CpGs hypomethylated in LGA (*n* = 36) compared to CTRL (*n* = 34, *p*-value < 0.001 and |methylation difference| > 25%; Figure 1A, Appendix A). This new set of DMCs was then used throughout the following analysis.

As the functional interpretation is performed at the gene level, each CpG (or DMC) must be linked to a specific gene. Thus, our ability to adequately infer the regulatory effect of a CpG and its target gene will affect our ability to identify relevant pathways. Standard analytical approaches usually rely on the distance between CpG and transcription start site (TSS) of the targeted gene and often only consider the top candidate DMC per gene, not taking into account the cell specific genomic context. Therefore, we refined the CpG-gene association to optimally assess the influence of DNA methylation changes on gene expression and enhance functional interpretation. We built a novel gene-methylation score considering (1) the distance between TSS and CpG; (2) the CpG overlap with expression quantitative trait loci (eQTL) annotation, as eQTL information allows us to identify tissue-specific genomic region links to gene expression changes; and (3) the regulatory annotation (e.g., Promoter, Enhancer) based on cell-specific histone marks [4] and on the Ensembl Regulatory database, as we know that the relationship between change in DNA methylation and change in gene expression will depend on a cell-specific genomic context (Figure 1B). We established 756,470 CpG-gene associations including 34% of them found through eQTL annotation. We then summarized the CpG information at the gene level, generating a gene-methylation score for each gene (*n* = 24,857, Appendix A). We first confirmed that the gene-methylation score properly recapitulates the influence of key parameters in DMC analysis such as significance and effect size of the methylation change, number of DMCs per gene, and distance from TSS, as well as promoter and enhancer localization (Figure 1C). We also confirmed that while preserving key information from standard methylation metrics, the gene-methylation score presented a better association with DEGs than significance or methylation change alone. Thus, the gene-methylation score appears to be a better predictor of the methylation influence on gene expression (Figure 1C). We then used our gene-methylation score to perform pathway enrichment analysis and data integration, especially considering integration with gene expression data.

Using the gene ontology (GO) reference database, we performed methylation gene-set enrichment analysis (GSEA) based on the gene-methylation score. We found that change in DNA methylation in LGA HSPC samples targeted genes involved in signaling regulating fetal development as well as in key stem cell pathways such as Wnt signaling, cell fate specification, and cell fate commitment pathways (adjusted *p*-value < 0.01, Figure 1D) confirming previous findings [3].

### 2.2. Single-Cell Transcriptomic Analysis Confirms Alteration of Hyper-Methylated Genes in Pathways Regulating Stem Cell Differentiation among LGA HSCs

To identify genes altered in LGA and to obtain further biological insight into the functional consequences of the DNA methylation modifications observed in LGA, we performed a single-cell transcriptomic analysis comparing CTRL and LGA HSPCs.

To enable lineage-specific transcriptomic analysis, we created a hematopoietic reference map (i.e., hematomap) by integrating data generated from cord blood-derived CD34+ HSPC cells (*n* = 18,520) from 7 control neonates (Figure 2A). Based on cluster-specific gene expression, we identified 18 distinct clusters representative of major lineages (Long-Term HSC, HSC, Multi-Potent progenitor, Lymphoid, Myeloid, and Erythroid) of the hematopoietic compartment (Figure 2B, Appendix A). Each cluster was annotated using cell-type-specific markers. Markers were then ranked based on their expression fold change and the specificity of the cluster. Top cluster-specific markers were compared with published cell-type-specific genes [5,6,9,10] (Appendix A). Candidate cell subpopulations were distributed as follows: 1% LT-HSC (*ID1*); 24% HSC (*AVP*); 45% MPP/LMPP (*CDK6*); Lymphoid (*CD99, LTB*); 1% B cell (*IGHM*); 1% T cell (*CD7*); 14% Erythro-Mas (*GATA1*); <1%Mk/Er (*PLEK, HBD*); 8% Myeloid (*MPO*); <1% DC (*CST3*, *CD83*).

To identify differentially expressed genes (DEG) between CTRL and LGA samples, we implemented the Hash Tag Oligonucleotide (HTO) multiplexing strategy [7] allowing simultaneous processing of CTRL and LGA samples. Multiplexing is a means to limit the influence of technique-driven batch effects at every stage of the analysis to improve the biological relevance of the finding. We generated multiplexed single-cell transcriptomic data from 6 LGA (*n* = 6861 cells) and 7 CTRL (*n* = 5823 cells) samples. In LGA samples, we observed a shift toward downregulated genes (Appendix A) especially in the HSC subpopulation (*n* = 285 downregulated genes over 373 DEGs, adjusted *p*-value < 0.05 and log_2_FC < (−0.5), Figure 2C; Appendix A). Notably, the well-known *EGR1*, *JUNB,* and *KLF2* genes were among the top affected genes. Using GO enrichment analysis, we found that downregulated genes were enriched in growth-related pathways (e.g., regulation of growth) as well as in stress-related biological processes (e.g., response to temperature stimulus, cellular response to chemical stress; Figure 2D, adjusted *p*-value < 0.05).

To assess if these HSC-specific transcriptomic changes may be associated with epigenetic changes, we integrated bulk DNA methylation with single-cell gene expression data using the gene-methylation score. We found that DEGs, and particularly the down-regulated genes, mostly showed high gene-methylation scores (Figure 3A,B). We then assessed the association between changes in DNA methylation and gene expression at the pathway level. We looked for enrichment for differentially methylated genes considering pathways that were identified based on DEGs. We found a significant overlap between GO terms enriched in LGA HSC downregulated genes and GO terms enriched in hypermethylated genes (10 out of 46; *p*-value < 0.05, hypergeometric test). The most co-enriched term is “regulation of growth” including notably *SOCS3*, *SIRT1* and *SESN2* genes that are both downregulated and within the top 10% of hypermethylated genes (Figure 3C). These results suggest that the epigenetic change in LGA could lead to an HSC-specific alteration of the regulation of growth signaling.

### 2.3. DNA Methylation Changes Occurs in HSCs and DEGs Associated Open Chromatin Regions

To assess if the HSC-specific transcriptional alteration could be due to HSC-specific epigenetic change, we profiled chromatin accessibility at the single-cell level (i.e., single-cell ATAC-seq). We generated open chromatin data across 8733 cells in HSPCs from 6 CTRL and 5 LGA neonates. We first annotated subpopulations using the label transfer approach between ATAC-seq data and the lineage labels from the Hematomap (Figure 4A, Supplemental Appendix A). To validate the relevance of our lineage annotation, we performed TF motif enrichment and observed that lineage-specific peaks were effectively associated with well-known lineage-specific TF (Supplemental Appendix A).

We then integrated our bulk DNA methylation data with our single-cell ATAC-seq data to assess DMCs distribution within open chromatin regions (OCRs). Overall, 31% of the 211,479 peaks contain CpGs queried by our genome-wide methylation assay. We first observed a strong enrichment for DMCs in OCRs with 74% of them located in OCRs compare to only 34% of overall queried CpGs (*p*-value < 0.001, hypergeometric test). Such enrichment further supports the putative regulatory influence of our DMCs. By performing lineage-specific analysis, we observed DMCs enrichment in HSC-specific open chromatin region with a total of 11% of HSC-specific peaks containing DMCs (adjusted *p*-value < 0.001, Figure 4B), while no enrichment was observed for the other lineages. This result corroborates the HSC-specific transcriptional impact of the DNA methylation changes observed in LGA. Furthermore, we observed that DEGs in LGA HSC and especially down-regulated genes were enriched for OCRs containing DMCs (Figure 4C).

Not limiting our analysis to the regulatory role of DMCs within open chromatin regions, we then assessed the change in chromatin accessibility in LGA HSCs. We identified 278 open chromatin regions that significantly differ between LGA and CTRL HSCs, with 215 showing decreased and 63 showing increased accessibility (adjusted *p*-value < 0.001 and |log_2_FC| > 0.25, Appendix A). By performing TF Motif analysis on regions with decreased accessibility, we identified that the motif of the transcriptionally downregulated TFs EGR1 and KLF2 are highly enriched (*p*-value < 1.10^−40^) and among the top 6 enriched motifs (Figure 4D, Appendix A).

We then assessed the interaction between DNA methylation, gene expression, and chromatin accessibility. Regions with decreased accessibility were also strongly enriched in peaks including DMCs and peaks associated with DEGs (Figure 4E), with 3-fold and a 2.5-fold enrichment, respectively. Furthermore, these regions were strongly enriched for peaks containing both DMCs and associated with DEGs (23-fold enrichment) illustrating that early epigenetic programming is actually not limited to changes in DNA methylation but also involves chromatin rearrangement targeting altered genes.

### 2.4. EGR1, KLF2, and KLF4 Are Key Upstream Regulators Influenced by Early Epigenetic Programming in LGA

To further characterize the molecular mechanisms affected in LGA HSCs and identified master regulators, we leveraged the single-cell resolution of our approaches to perform a co-regulatory network analysis. This approach allowed us to model the influence of upstream transcription factors (TF) on expression changes of downstream target genes. We performed co-expression analysis to identify genes co-regulated by the same TF, i.e., regulons, and filter each regulon based on the presence of TF motif within a cis-regulatory region (SCENIC). We identified a total of 250 regulons but only considered for further analyses the 106 regulons identified based on high confidence cis-regulatory motif. These regulons only rely on associations for which the presence of the TF motif was experimentally validated. We then scored the regulons activity in each cell using gene expression profiles of the entire regulons (AUCell). We observed that lineage-specific regulons are associated with concordant lineage determining hematopoietic TFs such as GATA2, GATA3, MEIS1, TAL1, TCF3, EGR1, CEBPB, HOXB4, SPI1, and STAT1/3 further supporting our subpopulation annotation and the SCENIC approach (Appendix A) [8].

To identify TF associated with the changes in gene expression observed in LGA HSC, we compared the regulon activity between CTRL and LGA. We found seven regulons with a significant decrease in activity in the LGA HSC population (adjusted *p*-value < 0.001 and |activity score fold change| > 10%, Appendix A). No regulons were upregulated. These regulons were associated with ARID5A, EGR1, KLF2, KLF4, KLF10, FOSB, and JUN (Figure 5A). Among them, ARID5A, EGR1, KLF2, FOSB, and JUN were part of the 10 top active regulons in HSCs (Appendix A). Based on functional enrichment analysis using as reference GO:BP gene sets, and HSC signatures of quiescence or proliferative state [11], we showed that these regulons were enriched in genes regulating stress response, proliferation, and HSC differentiation (Figure 5B).

To further support the association between change in DNA methylation and change in gene expression previously identified at the gene level, we performed GSEA analysis to identify regulon enriched for both differentially methylated and differentially expressed genes. We found 9 regulons enriched in both hypermethylated and downregulated genes (adjusted *p*-value < 0.01 and NES < −1.6), including the differentially active regulons ARID5A, EGR1, FOSB, JUN, KLF2, and KLF4 (Figure 5C). We also found 9 regulons enriched in hypermethylated and upregulated genes (adjusted *p*-value < 0.01 and NES > 1.6) with key HSPC-specific regulons such as SPI1 promoting myeloid differentiation; Ref. [12] and HOX family (HOXA9, HOXA10, HOXB4) promoting HSPC expansion (Figure 5C) [13,14,15].

To confirm the putative influence of methylation change on TF activity, we performed TF motif analysis considering the proximal regions surrounding each DMCs (±20 bp). We found significant enrichment for 23 TF motifs (adjusted *p*-value < 0.05, Figure 5D). Among them, we found EGR1 and several members of the Kruppel-like factors (KLF) family: KLF14, KLF5, KLF1, and KLF6. Furthermore, by taking advantage of our single-cell ATAC-seq data, we looked at the enrichment of the TF motif in open chromatin regions of HSC containing DMCs. We found a strong enrichment in EGR1, KLF2, and KLF4 motifs indicating that DNA methylation change occurred in active regions of the EGR1/KLF2/KLF4 TF network (Figure 5E).

### 2.5. Multimodal Co-Regulatory Network Recapitulating TF-Gene Interactions Influenced by Early Epigenetic Programming in LGA

Based on the integration of the DNA methylation, single-cell ATAC-seq, and single-cell RNA-seq data, we built a network recapitulating interaction between main TFs and downstream target genes within the principal regulons altered in LGA neonates: EGR1, KLF2, and KLF4 (Figure 6). EGR1, KLF2, and KLF4 regulons rely on highly interconnected (co-regulated) genes (Figure 6A). For each target gene, we confirmed the presence of a unique or shared upstream TF binding motif within the open chromatin regions. We observed a high concordance between the regulons and open chromatin motif analysis: 96%, 91%, and 95% of genes included in EGR1, KLF2, and KLF4 regulons, respectively, were associated with at least one peak containing the corresponding TF motif supporting the association between genes and TFs. We then looked for evidence of epigenetic modifications that may alter TF-target interactions. We annotated genes with associated open chromatin regions containing at least one DMC (middle area) or identified as differentially accessible between CTRL and LGA (inside area) (Figure 6B). Overall, 23% (*n* = 27) of genes targeted by these TFs networks have epigenetic alteration (DMCs or decrease accessibility) in open chromatin regions while 22% (*n* = 26 genes) appear downregulated in LGA. Finally, we highlight KLF2 as possible master regulators influenced by early programming. Indeed, we identified KLF2 as a hypermethylated and downregulated gene that interacts directly with EGR1 and KLF4 suggesting the downstream influence of KLF2 on these TFs. Conversely, KLF2 was not identified as part of EGR1 and KLF4 regulons suggesting that KLF2 is not a target of these TFs. This network also further validated *JUNB and SOCS3* being highly epigenetically altered in cis-regulatory regions (Figure 6C), as well as *ID1*, *CDKN1A*, *IER2*, *IER3*, and *IER5* as key downstream altered targets of KLF2, EGR1, and/or KLF4, again highlighting how early programming alters signaling involved in the regulation of cell proliferation and differentiation.

### 2.6. In Vitro Analysis Confirms the Alteration of HSPCs Differentiation Capacities in LGA

Our integrative analyses highlighted epigenetic and transcriptomic alterations targeting signaling pathways involved in the regulation of HSC differentiation and proliferation. Thus, we decided to challenge HSPC differentiation and proliferation potential in vitro using colony-forming unit (CFU) assays. After 14 days of expansion, colonies from 4 CTRL and 4 LGA samples were classified into three categories: those derived from common myeloid progenitors (CFU_GEMM), erythroid progenitors (BFU-E), and granulocyte-macrophage progenitors (CFU_GM) based on the morphology of each colony. We observed a significant decrease in the number of common myeloid progenitor colonies in LGA samples (*p*-value < 0.05; Figure 7A) as well as striking differences in shape and size of more differentiated colonies (Figure 7B). CFU_GEMM colonies are the product of a non-committed hematopoietic progenitor able to differentiate in both erythroid and myeloid lineage. In our samples, only HSC and MPP have these features, suggesting that the decreased CFU GEMM proportion in LGA reflects either fewer HSC/MPP in starting cell subpopulation composition or a decreased proliferation and differentiation capacity of these cells.

To evaluate these two possibilities, we monitored cell population distribution across conditions at molecular resolution using our single-cell expression dataset. We observed a decrease in HSC cells (*p*-value = 0.015) and a trend toward increased MPP cells (*p*-value = 0.13, Figure 7C) in LGA compared to CTRL. Another way to look at population shift is to use pseudotime, i.e., a measure that reflects how far an individual cell is in a differentiation process. Indeed, cord-blood-derived CD34+ HSPCs represent a heterogeneous population of cells ranging from progenitors to progressively restricted cells of the erythroid, myeloid, or lymphoid lineages as confirmed by our single-cell transcriptomic analysis. To follow cell distribution through these levels of differentiation and assess the influence of the LGA environment we used the pseudotime tool from Monocle [16] Collecting the pseudotimes across our different cell populations, we observed a positive correlation between pseudotime and lineage differentiation as expected (*r* = 0.99, Pearson correlation, Figure 7D, Appendix A). We then compared the distribution of the pseudotime between LGA and CTRL using the least differentiated cells as roots, i.e., the long-term HSCs. At the population level, we observed an increase in pseudotime in LGA (*p*-value < 0.001, Figure 7E). Indeed, we observed a decrease in the number of cells presenting pseudotime associated with the HSC state in LGA samples (*p*-value < 0.05) and a shift toward cells presenting elevated pseudotime suggesting that LGA HSCs exit quiescence and differentiate more quickly compared to CTRL HSCs (Figure 7E). Altogether, our analysis supports the association between LGA exposure and cell growth signaling targeted by DNA methylation and gene expression changes with alteration of differentiation and proliferation capacities.

## 3. Discussion

Here, we interrogated three major layers of the regulatory landscape in cord-blood-derived CD34+ HPSCs, DNA methylation, chromatin conformation, and gene expression. We characterized, in-depth and at single-cell resolution, the functional consequences associated with early DNA methylation changes observed in LGA neonates. Through, the integration of multiple datasets and the development of novel analytical approaches, we addressed a very challenging aspect of functional (epi)genomics, the interpretation of DNA methylation changes. Focusing on HSPCs, we believe that we contributed to a better understanding of how early environment shapes the hematopoietic compartment development and long-term function.

We demonstrated in LGA neonates a correlated increase in DNA methylation and change in chromatin accessibility associated with decreased expression of downstream target genes under the influence of key HSC transcription factors EGR1, KLF2, and KLF4. EGR1, KLF2, and KLF4 are zinc-finger transcription factors involved in HSC quiescence signaling. EGR1 has a known role in regulating cell growth, development, and stress response in many tissues. In HSPC, EGR1 plays a role in the homeostasis of HSCs regulating proliferation [17]. EGR1 promotes quiescence and decreases through differentiation. Interestingly, EGR1 has also been shown to interact with epigenetic regulators forming a complex with DNMT3 and HDAC1 [18] suggesting a possible role in the epigenetic remodeling observed in LGA HSC. The KLF family is implicated in key stem cell functions. KLF4 is the most well-known factor in this family due to its role in reprogramming somatic cells into induced pluripotent stem cells [19]. KLF4 has been identified as a target for PU.1 transcription factor required for lineage commitment in HSPCs [20]. KLF2 and KLF4 promote self-renewal in embryonic stem cells [21] but no study has looked specifically at KLF2 and KLF2/KLF4 interactions in HSPCs. Our data suggest direct and indirect (shared downstream target) interactions between these three transcription factors in HSPCs. EGR1, KLF2, and KLF4 represent targets to be further explored in order to challenge causality. Still, our findings lead to a better understanding of how early exposure can affect long-term hematopoietic maintenance in humans via epigenetic programming of the EGR1, KLF2, and KLF4 signaling. Furthermore, these coordinated epigenetic and transcriptomic changes target genes regulating growth signaling, such as *SOCS3*, *SIRT1,* and *SESN2* [22,23,24]. Alteration of growth signaling highlights the tight correlation between in utero environment and the epigenetic programming. Indeed, excessive fetal growth observed in LGA neonates results in part from gestational hyperglycemia, dyslipidemia, or over secretion of placental insulin-like growth factors [25,26,27]. Altogether, these results further illustrate how DNA methylation and chromatin accessibility are key co-epigenetics actors regulating TF activity. Such interplay was already observed in the context of lineage commitment [28,29], but not yet in the context of developmental programming of HSCs. This highlights the interest in considering both methylation and chromatin rearrangement in fetal programming studies to decipher putative epigenetic imprinting and functional consequences.

Interestingly, EGR1, KLF2, and KLF4 are not only involved in the regulation of proliferation and differentiation *per se* but are key factors of the immediate, early response involved in stimulation-related cell activation. *EGR1* and *KLF2* expression increase in response to extrinsic stimulation. Elevated *EGR1* and *KLF2* expression promote self-renewal and quiescence in HSC [17,21]. Our transcriptomic data suggests that such activation may be occurring in our samples with the activation of stress-related signaling. The primary scope of our study was not to characterize the environmental exposure that would trigger such responses. However, one can speculate that the activation could result from stress due to cold exposure or handling time inherent to sample preparation. Still, the decreased activity observed in LGA suggests that LGA HSCs’ capacities to respond to environmental challenges are diminished. This hypothesis fits with the concept of early programming in which disease susceptibility relies not only on early impairment of organ development but also on a decreased adaptability to further environmental challenges to trigger disease [30]. Indeed, fine-tuning HSC quiescence mechanisms is of crucial relevance for optimal hematopoiesis. Not responsive dormant HSC would lead to hematopoietic failure due to a lack of differentiated blood cells. Although highly responsive HSC would lead to exhaustion of the population and a lack of long-term maintenance of the hematopoietic system [31].

To validate findings from our integrative approach, we challenged HSPCs in vitro and found a significant decrease in the number of CFU-GEMM colonies, colonies containing both erythroid and myeloid cells. These colonies are likely to originate from HSC or MPP cells, as only these cells have this multi-potential. These alterations could result from the decreased differentiation and proliferation capacities of these CD34+ cells or a decrease in their initial proportion in LGA cord blood. Our data suggest that both are altered in LGA. Indeed, the cell population analysis at the transcriptomic level revealed a decrease in HSCs in LGA neonates but a tendency to an increase in MPPs. We also observed epigenomic and transcriptomic alterations in signaling pathways and transcription factors regulating differentiation and proliferations of HSCs. Yet, this loss of stemness capacities in HSC is likely to drive the decrease in HSC subpopulations observed in our data and the decreased colony-forming capacity.

These findings corroborate previous studies on the developmental programming of the hematopoietic system [7,8]. A reduction in self-renewal of HSPCs and increased differentiation in both lymphoid and myeloid lineages have been observed in a mouse model of maternal obesity [2]. These effects may drive long-term consequences in human health as illustrated by the study performed by Kotowski et al. in which the integrity of the hematopoietic system in neonates was associated with susceptibility to onset of hematopoietic pathologies [32].

Hematopoietic stem cell differentiation and self-renew rely on a synergic interplay between genetically encoded signaling, cell-intrinsic, and cell-extrinsic factors as well as epigenetic modifiers [33]. This interplay appears altered in LGA neonates. We here provide a comprehensive model recapitulating the functional influence of the epigenetic early programming on HSPCs fitness to later environmental exposure (Figure 7F). We also linked LGA-associated epigenetic modifications to gene expression and functional alterations through a novel integrative approach. In this regard, we identified targets to be further explored. We also brought a better understanding of how early exposure can affect long-term tissue maintenance via epigenetic programming of EGR1, KLF2, and KLF4 associated regulation of growth signaling.

## 4. Methods

See the Appendix A for additional information.

### 4.1. Clinical Sample Collection

Cord blood samples were obtained from CTRL and LGA neonates. LGA were defined by birth weight and ponderal index values greater than the 90th percentile for gestational age and sex. Control infants had normal parameters (between 10th and 90th percentiles) for both birth weight and ponderal index. Maternal and infant characteristics are shown in Appendix A.

### 4.2. Isolation of CD34+ HSPCs

Mononuclear cells were separated using PrepaCyte-WBC following which CD34+ cells were obtained by positive immunomagnetic bead selection, using the AutoMACS Separator (Miltenyi Biotech, Cologne, Germany). Cells were cryopreserved in 10% dimethyl sulfoxide using controlled rate freezing upon analysis.

### 4.3. Genome-Wide DNA Methylation Assay

DNA methylation levels for >1.7 M CpGs were obtained using the HELP-tagging assay as previously described [34].

### 4.4. Single-Cell RNA Sequencing Libraries Preparation

After cell count and viability check, the cell suspension was loaded into the Chromium controller (10× Genomics, Pleasanton, CA, USA) and library was generated using the chromium single-cell v3 chemistry following manufacturer recommendations. Gene expression library was sequenced using 100 bp paired-end reads on the Illumina NovaSeq 6000 system (Illumina, San Diego, CA, USA).

### 4.5. Single-Cell ATAC Sequen1cing Libraries Preparation

After cell count and viability check, nuclei were isolated from cell suspension and incubated with transposase. Transposed nuclei were then loaded into the Chromium 10× Genomics controller and library was generated using the chromium single-cell ATAC v1.1 chemistry following manufacturer recommendations. Gene expression library was sequenced using 150 bp paired-end reads on the Illumina NovaSeq 6000 system.

### 4.6. HTO Protocol

After cell counting and viability check and prior to cell suspension loading on the Chromium controller, cell hashtag (HTO) staining (Biolegend, San Diego, CA, USA) was used following the cell-hashing protocol [7].

### 4.7. Colony Forming Unit Assay

To assess clonogenic progenitor frequencies, 3 × 10^3^ CD34+ HSPC cells were plated in methylcellulose containing SCF, GM-CSF, IL-3, and EPO (H4434; STEMCELL Technologies, Vancouver, BC, Canada). Colonies were scored 14 days later.

### 4.8. Data Processing and Statistical Analysis

For DNA methylation analysis, low-quality CpGs were filtered out based on detection rate and confidence score. 754,931 out of 1,709,224 CpGs were conserved for further analysis. Linear regression and statistical modeling using the LIMMA R package [35] were used to identify differentially methylated CpGs (DMC) including maternal age, sex, ethnicity, batch, and library complexity in the linear model. We assessed enrichment for biological pathways performing GSEA using the ClusterProfiler package [36]. We performed transcription factor (TF) motif enrichment analysis using the HOMER tool [37] considering a 20 bp region around the DMCs.

For single-cell RNAseq (scRNA-seq) analysis, data were preprocessed using the CellRanger count pipeline (10× Genomics). Data filtering, normalization, and integration as well as cluster identifications were performed using Seurat (v4) pipeline [28]. Pseudo-bulk differential expression analysis between LGA and CTRL cells within each hematopoietic lineage was performed using DESeq2 R package including batches and sex of samples in the negative binomial model [29]. Over representation test was performed on differentially expressed genes (DEGs) using enrichGO and enrichKEGG of the ClusterProfiler Package. The SCENIC workflow [38] was used to identify co-regulated genes module associated to a TF (regulons) and to generate cell-specific activity scores for each regulon. Differentiation trajectory analysis and pseudotime attribution were conducted with Monocle [16].

For single-cell ATAC-seq, data were preprocessed using the CellRanger ATAC pipeline (10× Genomics). Data filtering, normalization, and integration as well as clustering were performed using the Signac pipeline. Cell type identification was based on scRNA-seq annotation using a label transfer approach. Peaks calling at lineage level was performed using the MACS2 tool. Peaks specific to each lineage or differentially accessible between LGA and Control were identified using the FindMarkers function with Logistic Regression (LR) models including cellular sequencing depth as a latent variable. TF motif enrichment on lineage or group-specific peaks was performed using the FindMotifs function. All peaks enrichment analysis was performed using hypergeometric tests. For final Gene Regulatory Network (GRN) construction, TF target interactions inferred with SCENIC were filtered out based on the presence of a corresponding TF motif in the peak associated with the target. Appendix A contains information on the number of cells per sample.

### 4.9. Gene-Methylation Score

To compute the gene-methylation score, 2 steps were needed: (1) to generate a CpG score that reflects the association between CpG and gene, and (2) to concatenate CpG-scores at the gene level.
(1)CpG-score
CpGScore = (−log10(p_cpg_) × meth.change) × LinkWeight × RegWeight
where p_cpg_ is the nominal *p*-value of the differential methylation analysis, and meth.change is the difference between the percentage of methylation in LGA and the percentage of methylation in CTRL. LinkWeight represents the confidence in CpG-gene association and RegWeight represents the estimated regulatory influence of the considered CpG based on CD34+ specific genomic annotation defined using CD34+ specific histone marks as previously described [3] and EnsRegScore refers to regulatory regions defined based on the Ensembl Regulatory build hg19 genome annotation [39].
(2)To concatenate CpG-Scores at gene level: gene-methylation score

To summarize the CpG methylation change at the gene level, we aggregated the CpG-Scores into a methylation gene score by taking care to (i) alleviate the arbitrary number of CpGs per gene and (ii) interpret differently CpG influences located on the promoter of them in others genomic region.

The gene-methylation score is defined as:

Gene-methylation score = ($\sum CpG\;Score\times Weight{\;}_{n\_cpg}$)_promoter_ + ($\sum \left|CpG\;Score\right|\times Weight{\;}_{n\_cpg}$)_other_regions_

Where the *Weight_nCpG_* was optimized to alleviate the influence of the number of CpGs linked to a gene and defined as:$${Weight}_{nCpG}=\sqrt[{3,8}]{\frac{1}{\sum \frac{1}{\left|CpGScore+1\right|}}}$$

The code to perform the analyses in this manuscript is available at https://github.com/umr1283/LGA_HSPC_PAPER.git (last accession date: 29 May 2022).

## Figures and Tables

**Figure 1 ijms-23-07323-f001:**
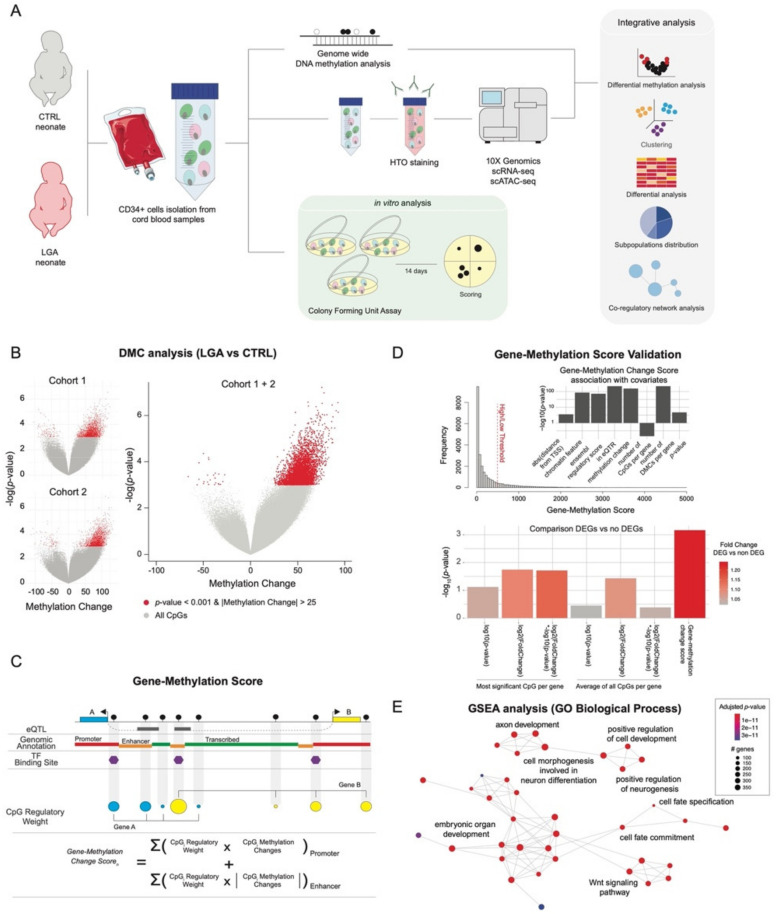
**LGA is associated with DNA hypermethylation targeting key stem cell signaling pathways.** (**A**) Overview of study design (**B**) Volcano plot of DNA methylation score differences for LGA compared to CTRL in cohort 1, cohort 2, and cohort 1 + 2. Differentially methylated loci with *p*-value < 0.001 and |methylation difference| > 25% are shown in red. (**C**) Summary of calculation for the gene-methylation score. (**D**) Validation of the gene-methylation score. Gene-methylation score distribution. Bar plot of the association between gene-methylation score and genomic or methylation-related features. Bar plots representing the significance of the difference in gene-methylation score of DEGs compared to non-DEGs considering different metrics. eQTR, region with expression quantitative traits loci; DMC, differentially methylated CpGs. (**E**) Network representation of GO Biological Process enriched in hypermethylated genes. Significantly enriched GO terms were identified using GSEA based on the gene-methylation score. Edges represent interactions (gene overlap) between pathways.

**Figure 2 ijms-23-07323-f002:**
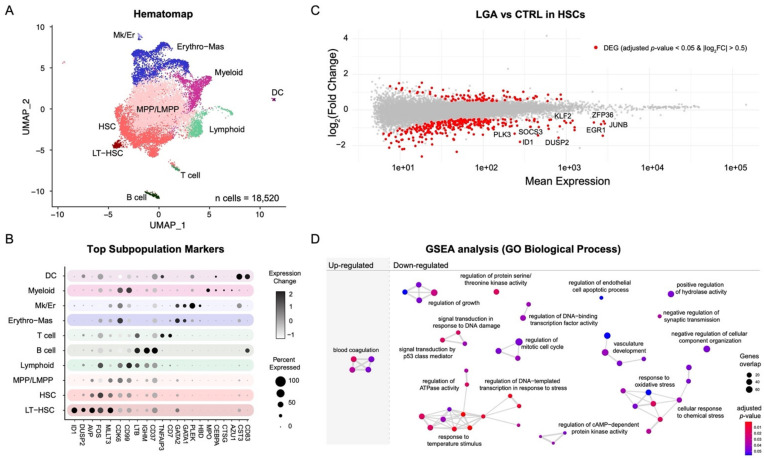
**Lineage-specific transcriptomic analysis.** (**A**) Hematomap, UMAP representation of distinct HSPC lineages. (**B**) Dot plot representing key markers used to annotate cell populations. LT-HSC, long-term hematopoietic stem cell; HSC, hematopoietic stem cell; MPP, multipotent progenitor; LMPP, lymphoid-primed multipotent progenitors; Erythro-Mas, erythroid and mast precursor; Mk/Er, megakaryocyte and erythrocyte; DC, dendritic cell. (**C**) MA plots representing gene expression analysis in HSCs comparing LGA vs. CTRL. Differentially expressed genes with adjusted *p*-value < 0.05 and |log_2_FC| > 0.5 are shown in red. (**D**) Network representation of significantly enriched pathways identified through GO GSEA analysis comparing LGA vs. CTRL. Non-redundant pathway annotations have been used. Edges represent interactions between pathways.

**Figure 3 ijms-23-07323-f003:**
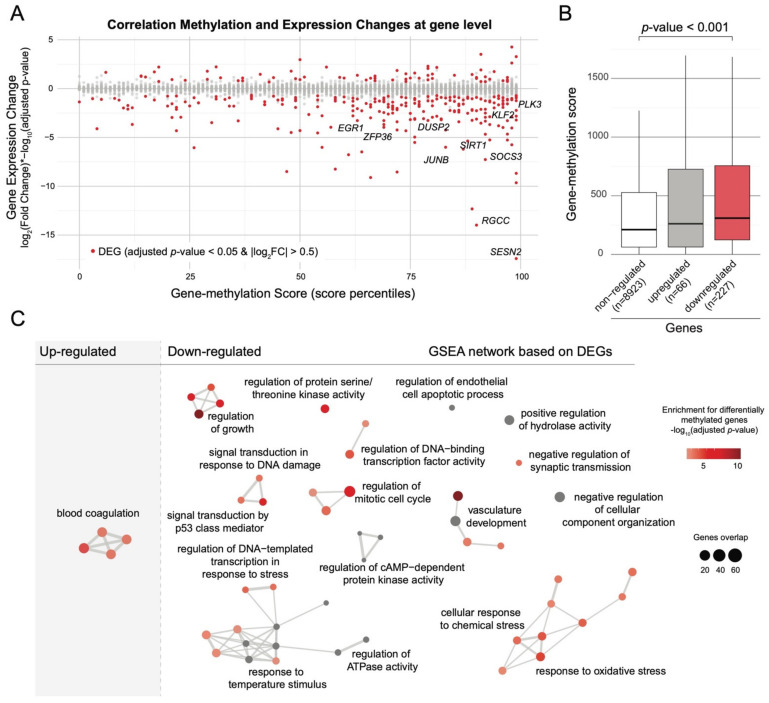
**Association between changes in DNA methylation and in gene expression.** (**A**) Dot plot representing the correlation between DNA methylation and gene expression changes. Differentially expressed genes with adjusted *p*-value < 0.05 and |log_2_FC| > 0.5 are shown in red. (**B**) Boxplots representing gene-methylation score distribution associated with non-DEG, up-regulated, and down-regulated genes (Wilcoxon test). (**C**) Network representation of significantly enriched pathways identified through GO GSEA analysis based on DEG identified comparing LGA vs. CTRL. Nodes are color-coded based on enrichment for differentially methylated genes using the gene-methylation score. Edges represent interactions between pathways.

**Figure 4 ijms-23-07323-f004:**
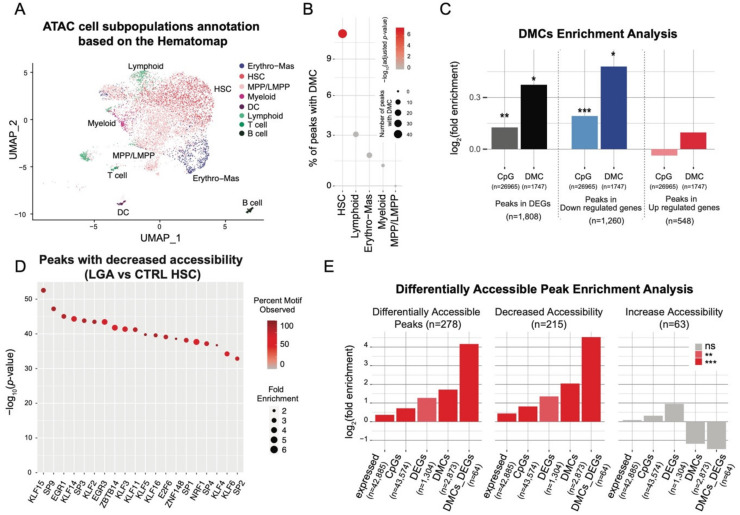
**Chromatin accessibility analysis.** (**A**) UMAP representing HSPCs lineage based on chromatin accessibility. Annotations are based on the Hematomap using the transfer label approach. (**B**) Dot plot representing enrichment for DMC within lineage-specific peaks. (**C**) Bar plots representing enrichment for peaks containing CpG or peaks containing DMC associated to DEGs, up-regulated and down-regulated genes (* *p*-value < 0.05; ** *p*-value < 0.01, *** *p*-value < 0.001, hypergeometric test). (**D**) Dot plot representing enrichment for transcription factor motif within Down peaks identified comparing chromatin accessibility between LGA and CTRL. Dots are color-coded based on percentage of peaks with motif and y-axis represents the significance of the enrichment. (**E**) Bar plots representing enrichment analysis considering accessible, down, and up peaks. Enrichment is performed using peaks in expressed genes (expressed), peaks with CpGs (CpGs), peaks in DEG (DEGs), peaks with DMC (DMCs), and peaks in DEG with DMC (DMCs_DEGs) as reference gene sets (** *p*-value < 0.01, *** *p*-value < 0.001, hypergeometric test).

**Figure 5 ijms-23-07323-f005:**
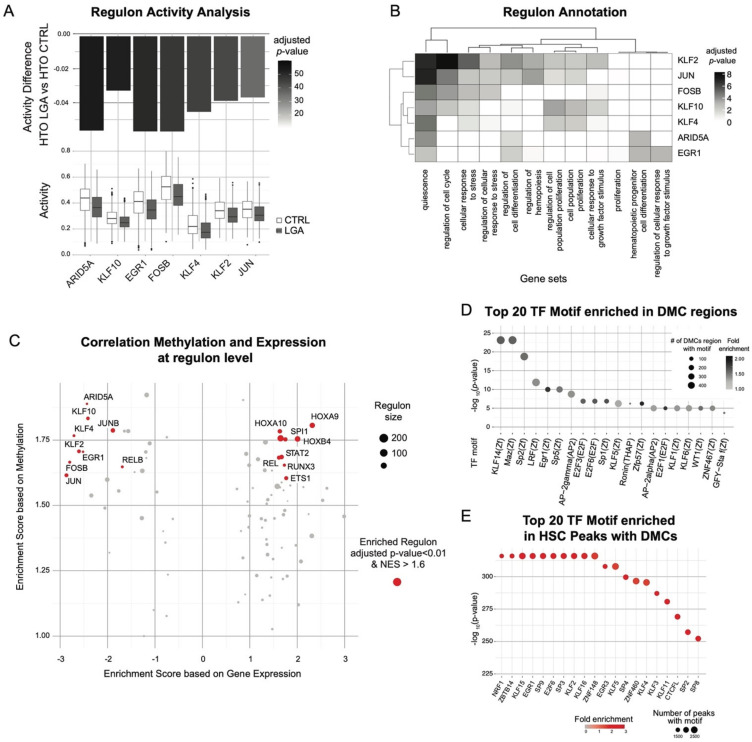
**Epigenetic programming of HSC-specific regulons altered in LGA neonates.** Regulons and TF target information were obtained through the SCENIC workflow. (**A**) Boxplots representing regulon activity score in CTRL and LGA HSC lineage. Barplot representing the change in regulon activity and significance comparing LGA vs. CTRL. Only significantly affected regulons are represented (adjusted *p*-value < 0.001 and |activity score fold change| > 10%). (**B**) Heatmap representing association between altered regulons and selected gene sets annotation. (**C**) Volcano plot representing enrichment in the change in expression and DNA methylation in regulons. Regulons enriched considering both expression and methylation (adjusted *p*-value < 0.01 and NES > 1.6) are in red. (**D**) Dot plot representing enrichment for TF binding motifs using HOMER considering a ±20 bp region around DMCs. Dots are color-coded based on the significance of the enrichment and y-axis represent the number of regions with binding motif among DMCs. (**E**) Dot plot representing enrichment for TF binding motifs using HOMER considering peaks with DMCs. Dots are color-coded based on the fold-enrichment and y-axis represents the significance of the enrichment.

**Figure 6 ijms-23-07323-f006:**
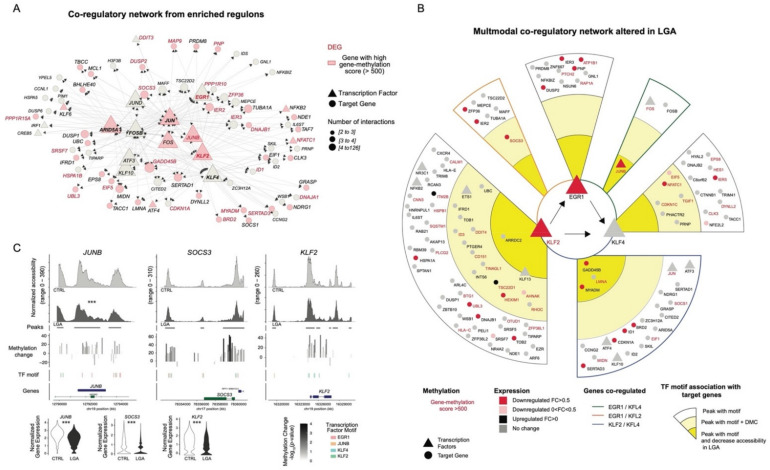
**Network recapitulating interaction between the epigenomic and transcriptomic alterations in LGA.** (**A**) Network representing interactions between target genes and transcription factors considering our top affected regulons ARID5A, EGR1, KLF2, KLF4, FOSB, and JUN. Each dot represents a gene within the network, the triangle represents a transcription factor, the arrow represents the interaction between the transcription factor and target genes, shapes are color-coded to reflect the change in gene-methylation score, and DEGs are labeled in red. Size of the shape represents the number of interactions. Only genes with two or more interactions are represented. (**B**) Tracks representing DNA methylation and chromatin accessibility for selected representative regions. Histogram representing change in DNA methylation at CpG level comparing LGA vs CTRL. Violin plot representing gene expression for selected genes (**C**) Network-based on the integration of DNA methylation, gene expression, and chromatin accessibility representing transcription factors and downstream target interactions within EGR1, KLF2, and KLF4 regulons. Only genes associated with peaks with TF motifs of interest are annotated. Donuts represent different levels of interactions. ***: significant change of peak accessibility (logistic regression) or gene expression (Wilcoxon test) in LGA compared to Control HSCs, adjusted *p*-value < 0.001.

**Figure 7 ijms-23-07323-f007:**
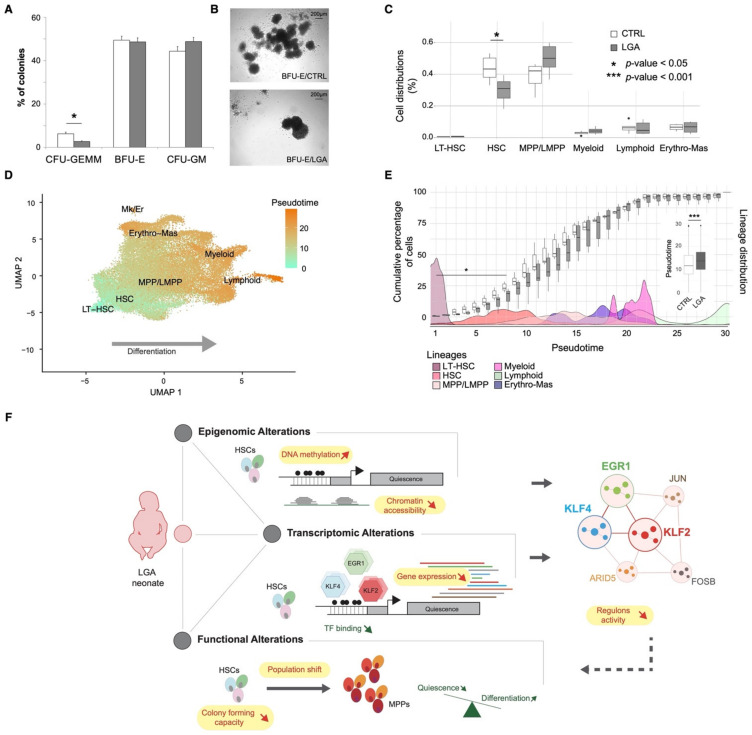
**LGA is associated with decreased expansion capacity and an HSC shift toward more differentiated cells.** (**A**) Bar plot representing colonies distribution after CFU assays. (**B**) Representative capture of colonies’ morphological differences found in CTRL and LGA. (**C**) Boxplots representing the cell distribution across hematopoietic main lineages in CTRL and LGA. (**D**) UMAPs representing pseudotimes across lineages. (**E**) Box plots representing the cumulative percentage of cells per pseudotime in CTRL and LGA. Boxplots in the vignette represent overall pseudotime distribution in CTRL and LGA. Density plots correspond to cell populations distribution across pseudotimes. (**F**) Model recapitulating the influence of LGA on the hematopoietic compartment. (LT-HSC, long-term hematopoietic stem cell; HSC, hematopoietic stem cell; MPP, multipotent progenitor; LMPP, lymphoid-primed multipotent progenitors; Erythro-Mas, erythroid and mast precursor; Mk/Er, megakaryocyte and erythrocyte; DC, dendritic cell; CFU-GEMM, common myeloid progenitors; BFU-E, erythroid progenitors; CFU-GM, granulocyte-macrophage progenitors.

## Data Availability

The DNA methylation and gene expression data will be made available upon request to A.P., P.F. or F.D.

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
