# Peer review of "Epigenetic and Transcriptomic Programming of HSC Quiescence Signaling in Large for Gestational Age Neonates"

_ijms, 2022, doi:10.3390/ijms23137323_

Round 1

Reviewer 1 Report

The authors of the manuscript titled, “Epigenetic and transcriptomic programming of HSC quiescence signaling in large for gestational age neonates” have used single-cell epigenomics, single-cell transcriptomics, and in vitro analyses to show functional link between DNA methylation changes to putative alterations of HSPC functions.  They showed DNA hypermethylation and chromatin rearrangement target a specific network of transcription factors that are known to sustain stem cell quiescence. They have also found a decrease in the expression of 26 key genes regulating HSC differentiation that includes EGR1, KLF2, SOCS3, and JUNB.  Authors have further reported that epigenetic programming is associated with a decreased ability for HSCs to stay quiescent. The manuscript is well written and is an important contribution for the advancement of the field. Overall, the introduction, materials and methods, results are well presented but I do suggest in the discussion section, the results should be discussed in more detail and should be supported by previous literature.

Author Response

We thank the reviewer for his time and his comments. We went through the manuscript to carefully check for spelling errors.

We made changes to the discussion by adding few references and by discussing target genes, not only focusing on the identified TFs, in the context of excessive fetal growth and long-term consequences, as suggested by the reviewer.

We believe that we addressed the reviewer's comments and we want to thank him again as they helped improve the impact of our manuscript.  

Reviewer 2 Report

Dear authors you manuscript entitled "Epigenetic and transcriptomic programming of HSC quiescence signaling in large for gestational age neonates" it is really interesting manuscript for the field.

Only few things regarding the english quality and figures could improve the high quality of scientific information.

So, please check carefully all the manuscript, revising each section, including abstract; for example: change to stay for remain, and so on.

Regarding the quality of figures, I could not see very well the words inside the figure, please correct.

Furthermore, consider to adding some experimental scheme figure as a first figure in order to understand better each procedure.

Thank you so much.

Best regards,

Author Response

We would like to thank the reviewer for his time and his relevant comments. 

Regarding the quality of the english, we carefully check for spelling errors throughout the manuscript and rephrase when appropriate. 

For the quality of the figures, we increased the font size in the different panels and the resolution to improve word visibility. We also added in Figure 1 an overview of the study design to help with the understanding of each experimental procedure as suggested by the reviewer. 

We believe by doing so we addressed the reviewer's comments and improved the overall quality of the manuscript.